# *CDKN2A* Gene Mutations: Implications for Hereditary Cancer Syndromes

**DOI:** 10.3390/biomedicines11123343

**Published:** 2023-12-18

**Authors:** Anastasiia Danishevich, Airat Bilyalov, Sergey Nikolaev, Nodirbec Khalikov, Daria Isaeva, Yuliya Levina, Maria Makarova, Marina Nemtsova, Denis Chernevskiy, Olesya Sagaydak, Elena Baranova, Maria Vorontsova, Mariya Byakhova, Anna Semenova, Vsevolod Galkin, Igor Khatkov, Saida Gadzhieva, Natalia Bodunova

**Affiliations:** 1SBHI Moscow Clinical Scientific Center Named after Loginov MHD, 111123 Moscow, Russiadr.onco.kh@gmail.com (N.K.); d.isaeva@mknc.ru (D.I.);; 2Institute of Fundamental Medicine and Biology, Kazan Federal University, 420008 Kazan, Russia; 3LLC Evogen, 115191 Moscow, Russia; 4Federal State Budgetary Institution Russian Scientific Center of Roentgenoradiology, Ministry of Healthcare of the Russian Federation, 117997 Moscow, Russia; 5Research Centre for Medical Genetics of N.P. Bochkov, 115522 Moscow, Russia; 6Federal State Autonomous Educational Institution of Higher Education I.M. Sechenov, Ministry of Health of Russian Federation, 119991 Moscow, Russia; 7FSBEI HE “Privolzhsky Research Medical University”, Ministry of Health of Russian Federation, 603950 Nizhny Novgorod, Russia; 8Federal State Budgetary Institution National Medical Research Center of Cardiology Named after Academician E.I. Chazov, Ministry of Health of the Russian Federation, 121552 Moscow, Russia; 9Russian Medical Academy of Continuous Professional Education, Russia, 125993 Moscow, Russia; 10Faculty of Medicine, Lomonosov Moscow State University, 119991 Moscow, Russia; 11The National Medical Research Center for Endocrinology, 117292 Moscow, Russia; 12Moscow Healthcare Department, Moscow State Budgetary Healthcare Institution Moscow City Oncological Hospital No. 1, 117152 Moscow, Russia; 13Moscow Department of Healthcare, 127006 Moscow, Russia

**Keywords:** *CDKN2A*, melanoma, pancreatic cancer

## Abstract

Malignant neoplasms, including pancreatic cancer and melanoma, are major global health challenges. This study investigates melanoma pancreatic syndrome, a rare hereditary tumor syndrome associated with *CDKN2A* gene mutations. *CDKN2A* mutations contribute to a lifetime risk of melanoma ranging from 28% to 67%. This study reports the clinical features of six individuals with *CDKN2A* mutations and identifies recurrent alterations such as c.307_308del, c.159G>C and c.71G>C. It highlights the need for *CDKN2A* mutation testing in suspected cases of familial atypical multiple mole melanoma. Clinically significant variants show associations with melanoma and pancreatic cancer. The challenges of treating individuals with *CDKN2A* mutations are discussed, and the lack of specific targeted therapies is highlighted. Preclinical studies suggest a potential benefit of CDK4/6 inhibitors, although clinical trials show mixed results. This study underscores the importance of continued research into improved diagnostic and therapeutic strategies to address the complexities of hereditary cancer syndromes.

## 1. Introduction

Malignant neoplasms are the second leading cause of death worldwide behind cardiovascular disease [1,2]. Notably, pancreatic cancer and melanoma emerge as two of the most aggressive malignancies [3]. As we delve into the pharmaco-economical aspects, the economic considerations surrounding genetic testing, counseling, and potential targeted therapies underscore the need for cost-effective strategies to address the hereditary nature of these malignancies, while ensuring accessibility and sustainability in healthcare delivery [4]. In addition, a number of genes have been associated with a high risk of hereditary cancer development [5]. One of those rare hereditary tumor syndromes is melanoma pancreatic syndrome (MPS) [6].

The first case of familial melanoma was presented by the physician Norris in 1820. His patient, a 59-year-old man, had a confirmed diagnosis of melanoma. This patient had a phenotypic manifestation of a large number of nevi throughout the body, coupled with a family history of melanoma [7]. A century after this case, Lynch and Krush elucidated the characteristics of MPS, which constituted a hereditarily linked ailment associated with an elevated susceptibility to the development of malignant neoplasms of the pancreas [8].

Melanoma pancreatic syndrome or familial atypical multiple mole melanoma (FAMMM) (OMIM# 606719) is a hereditary tumor syndrome characterized by an increased risk of melanoma and pancreatic cancer in individuals with a germline mutation in the *CDKN2A* gene [9]. Generally, pathogenic mutations in the *CDKN2A* gene are linked to a lifetime melanoma risk ranging from 28% to 67% before the age of 80. These risks are based on geographical region; there is a 1.9% risk in the general population [10]. For instance, a study conducted by Lynch H.T. et al. in 2008 showed data indicating a 13–22-fold elevation in the risk of pancreatic cancer development in individuals with familial melanoma, and those with p16/CDKN2A mutations exhibited a 38-fold increased risk of pancreatic cancer compared to the general population [11]. In some families with MPS, the *CDKN2A* mutation is associated with an elevated risk of pancreatic and duodenal carcinoma [12]. In addition, mutations in this gene are associated with melanoma–astrocytoma syndrome (OMIM# 155755).

The *CDKN2A* gene is classified within the family of tumor suppressor genes and is categorized as a “watchman of the cell cycle.” It encodes two distinct proteins, exerting its effect on the signaling pathways responsible for cell cycle regulation [13,14]. As shown in Figure 1, CDKN2A consists of four exons—1α, 1β, 2, and 3—resulting in the production of two distinct proteins through alternatively spliced transcripts. Exons 1α, 2, and 3 contribute to the synthesis of the p16 protein, which binds to cyclin-dependent kinase 4 (CDK4) and inhibits its interaction with cyclin D [15]. This inhibition of interaction prevents the phosphorylation of the retinoblastoma protein and subsequently prevents the transition from the G1 to S phase of the cell cycle. Exons 1β, 2, and 3 encode the p14ARF (p14) protein, which functions in stabilizing p53 by inhibiting MDM2-induced p53 degradation [16]. CDKN2A mutations can disrupt the regulation of the cell cycle, resulting in the uncontrolled proliferation of abnormal cells [17]. Further investigations have revealed that mutations impacting p16 are predominantly characterized by loss-of-function missense mutations, although cases of nonsense mutations, promoter mutations, frameshift mutations, in-frame insertions, splicing alterations, and gene deletions have also been documented [18]. Conversely, mutations that disable p14 typically include splice site variations, inactivating frameshift mutations, or deletions specific to exon 1β [19,20].

The prevalence of *CDKN2A* gene germline mutations in the general population has not been determined. In the Italian population study by Ghiorzo, *CDKN2A* germline mutations were identified in 13 (5.7%) of 225 Italian patients diagnosed with pancreatic cancer [21]. Among the 16 probands with a familial history of cancer, including cases of pancreatic cancer and melanoma, clinically significant genetic variants of *CDKN2A* were identified in five (31%) cases [21]. Specific clinical indicators of this syndrome encompass the presence of multiple melanocytic nevi exhibiting signs of dysplasia, along with documented cases of melanoma in the family history. Furthermore, variants of the *CDKN2A* gene could be correlated with coronary heart disease, myocardial infarction, and atherosclerosis [22].

*CDKN2A* mutation testing is a crucial diagnostic tool in the assessment of individuals with suspected FAMMM syndrome. Even though FAMMM syndrome is rare, it is possible to suspect it and propose genetic testing of the *CDKN2A* gene if a patient fulfills diagnostic criteria that consist of the following clinical features [23]:High total nevi count: individuals typically exhibit a high number of nevi, often exceeding 50 in total.Specific nevi histologies: nevi in affected individuals may present with specific histological patterns, such as a lentiginous pattern and nuclear atypia.Presence of melanoma cases in relatives: a key criterion involves the document melanoma cases in at least one first- or second-degree relative.

Given these criteria, genetic counseling referral is strongly recommended in cases where individuals or families meet certain criteria [24]:Three or more invasive cutaneous melanomas: individuals with a personal history of three or more invasive cutaneous melanomas should be considered for p16/CDKN2A genetic testing.Mixed cancer diagnoses: cases involving a combination of invasive melanoma, pancreatic cancer, and/or astrocytoma diagnoses in an individual or within a family should be referred for genetic testing.

Thus far, there are no distinct guidelines provided for the surveillance and screening of melanoma pancreatic syndrome. Table 1 outlines the recommendations for monitoring patients with hereditary pancreatic cancer and melanoma, as per the National Comprehensive Cancer Network (NCCN) guidelines.

The aim of this study is to provide a description of the genetic and clinical characteristics observed in carriers of *CDKN2A* genetic variants diagnosed with malignant neoplasms.

## 2. Materials and Methods

The article outlines the clinical characteristics displayed by six patients who had CDKN2A mutations and were examined or treated for malignant neoplasms of the different localization at the Loginov Moscow Clinical Scientific Center and City Clinical Oncological Hospital No. 1 (Moscow) from 2020 to 2023. The mean age of the patients described in the article was 48.5 ± 12.5 years. Two male and four female patients were included in the study. All patients were consulted by a geneticist, provided information about their personal and family history of cancer, and signed an informed consent form. The form included information about molecular genetic testing and permission to use their depersonalized data for research and scientific publications.

A search for mutations in the CDKN2A gene was performed using either the targeted NGS panel or WGS. Targeted NGS testing of 111 cancer-associated genes was performed in the Laboratory of Oncogenetics and Hereditary Diseases in the Loginov Moscow Clinical Scientific Center, as previously described [25]. WGS was performed in a large-scale scientific research project that was implemented in Moscow (Russia) between 2021 and 2023 with the aim of identifying hereditary cancer syndromes (HCS) using WGS in the patients with colorectal, breast, pancreatic, ovarian, gastric, and endometrial cancer, as well as neuroendocrine tumors in the LLC Evogen, as previously described [26].

### 2.1. Targeted Sequencing

The generation of sequencing libraries was performed using KAPA HyperPlus kit (Roche, Basel, Switzerland). The DNA was fragmented either enzymatically or through ultrasonication, as per the guidelines provided by the manufacturer. The fragment sizes within the libraries were then assessed using the Agilent 2100 Bioanalyzer (Agilent technologies, Santa Clara, CA, USA).

For a quantitative evaluation of the final libraries, the Qubit 3 fluorometer (Thermo Fisher Scientific, Waltham, MA, USA) was employed. This step was crucial to confirm that each library was pooled in equal molar amounts for subsequent sequencing processes. To enrich the libraries for specific genes, the NimbleGen SeqCap EZ Choice kit by Roche was used.

The sequencing itself was carried out using the Illumina MiSeq platform, along with the MiSeq Reagent Kit v2 (500 cycles). This kit is a product of Illumina, based in San Diego, CA, USA. To ensure quality control in terms of cluster generation, sequencing, and alignment, a control library, specifically the PhiX Control v3, was incorporated into the process.

### 2.2. Whole Genome Sequencing

WGS was performed using NGS sequencers, DNBseq-T7 and DNBseq-G400 (MGI, China), using a PCR-free enzymatic shearing protocol for library preparation (MGIEasy FS PCR-Free DNA Library Prep Kit (MGI, Shenzhen, China)). All the subsequent stages, including PE150 (paired-end 2 × 150 bp) sequencing, were carried out in accordance with the manufacturer’s standard protocols. The average sequencing depth was 30×. The identification of the genetic variants was carried out using bioinformatics analysis accelerators (EVA Pro, EVOGEN, Moscow, Russia; MegaBOLT, MGI, Shenzhen, China). During the analysis of WGS results, special expert attention was carried out primarily for cancer-associated genes. The gnomAD database was used to estimate the population frequencies of the identified variants.

For the WGS results, Sanger validation (*CDKN2A* variants) was performed in the EVOGEN LLC. Sample DNA extraction was performed using QIAamp DNA blood Mini Kit (QIAGEN, Hilden, Germany)/MGIEasy Magnetic Beads (MGI, Shenzhen, China) by standard manufacturer protocols. Purified PCR products were sequenced using the BigDye Terminator Kit and ABI 3500 Genetic Analyzer (Applied Biosystems, Waltham, MA, USA) by standard manufacturer’s protocols. The analysis of the sequenced data was performed using the software Variant Reporter Software v3.0 (Applied Biosystems, Waltham, MA, USA).

## 3. Results

In our study, we report the clinical features of FAMMM in six individuals with pathogenic mutations in the *CDKN2A* gene. Out of those six patients, four were diagnosed with pancreatic cancer or melanoma, one patient had a benign tumor status and another one was characterized by multiple cancer localizations. The mean age at which pancreatic cancer occurred was 50 ± 14 years. Melanoma was diagnosed in a 36-year-old patient. There were two patients diagnosed with multiple primary tumors. The age of onset for the first cancer was 36 and 39 years of age.

In three out of six cases missense variants were identified and then three out of six patients had deleterious variants. In the majority of our patients, we identified recurrent alterations in the *CDKN2A* gene. The c.307_308del variant was found in three out of six patients, two of whom had pancreatic cancer and one of whom had multiple primary tumors (colorectal and biliary tracts cancers). The c.71G>C variant was identified in two out of six cases: one in a patient with pancreatic cancer and the other in a proband with multiple dysplastic nevi. The c.159G>C was diagnosed in one out of six patients with pancreatic neuroendocrine tumors and melanoma (Table 2).

All patients included in the study had a family history of cancer. Specifically, three individuals had first-degree relatives with cancer and three others had second-degree relatives with mutations. 

## 4. Discussion

Studying mutations in the *CDKN2A* gene is key to understanding the complex nature of cancer development. *CDKN2A* functions as a tumor suppressor gene that controls the cell cycle through the production of p16INK4A (p16) and p14ARF (p14) proteins, and thus plays a critical role in protecting cellular integrity [16]. The study of CDKN2A mutations provides a valuable insight into the molecular abnormalities that drive carcinogenesis.

Understanding the spectrum of mutations affecting *CDKN2A* is critical to identifying their role in cancer susceptibility and progression. Mutations that disrupt the function of p16, often presented by missense mutations resulting in a loss of function, can disrupt the delicate balance of cell cycle control [18]. Nonsense mutations, frameshift mutations, in-frame insertions, splicing irregularities, promoter alterations, and gene deletions contribute to the various mutational patterns associated with p16.

Given the complex interplay between mutations in *CDKN2A* and dysregulated cell cycle control, the study of these genetic abnormalities offers great potential for the development of targeted therapeutic strategies. By shedding light on the particular mutations in *CDKN2A* that are associated with different types of cancer, researchers and clinicians can lay the groundwork for precision medicine approaches that tailor treatments to the unique genetic profiles of individual patients. Ultimately, the study of *CDKN2A* mutations not only enhances our basic understanding of cancer biology, but also provides concrete opportunities to advance personalized cancer therapy.

The Clinvar Database has registered a total of 1324 clinically significant variants of *CDKN2A* gene mutations, with 183 of them classified as pathogenic [27]. The majority of these pathogenic mutations manifest as duplications, deletions, or single-nucleotide substitutions.

In our study, we identified a total of three clinically significant variants in the *CDKN2A* gene: c.307_308del, c.71G>C, and c.159G>C.

The variant c.159G>C results in the substitution of methionine to isoleucine at the 53rd codon of the CDKN2A (p16INK4a) protein (p.Met53Ile). These amino acids share similar properties. The pathogenic mutation p.M53I (also denoted as c.159G>C) is situated in the coding exon 2 of the *CDKN2A* gene. It arises due to the replacement of G to C at position 159 of the nucleotide. According to the VarSome database, this variant was presented in 28 publications. This alteration is most frequently observed in individuals of Scottish descent [28,29]. It has also been documented in cases of familial melanoma. In a Norwegian study, the variant C.159G>C was identified in several probands with melanoma and pancreatic cancer [30]. Experimental studies have shown that this missense change affects CDKN2A (p16INK4a) function [29,31,32]. For example, Becker T.M. et al. reported that the cell-cycle-inhibitory activity of mutants with p.M53I variants was profoundly reduced, and the partially retained capacity for CDK4 binding in functional assays did not correlate with the significant preservation of cell-cycle-regulatory function [32].

The pathogenic mutation p.R24P (also known as c.71G>C) is located in the first coding exon of the *CDKN2A* gene. It is characterized by a G to C nucleotide replacement at position 71. Variant c.71G>C is a missense mutation, leading to the substitution of arginine to proline at the 24th codon of the CDKN2A protein (P16INK4A). Prediction tools indicate that this variant does not impact RNA splicing itself. Functional studies have demonstrated that c.71G>C results in the loss of binding to the CDK4/6 receptor, consequently disrupting the mechanism of cell cycle inhibition. This variant has been identified in over 40 melanoma and pancreatic cancer patients [33,34].

The variant c.307_308delCG is characterized by the deletion of two nucleotides (C/G), causing a reading frame shift. This, in turn, leads to an arginine to alanine replacement at codon 103, and a premature stop codon is introduced at the 16th position of the reading frame. Consequently, either protein truncation or mediated mRNA decay occurs, resulting in the abnormal functioning of p16INK4a and p14ARF. According to the VarSome database, the variant chr9:21971051delCG has been described in four articles on melanoma and pancreatic cancer. The study identified the occurrence of the c.307_308 delCG variant in three out of six patients, regardless of the malignant neoplasm location. One patient had primary multiple manifestations of colorectal cancer and biliary tract cancer. In this case, the patient had a burdened family history of cancer. Her mother was diagnosed with breast cancer at 50 years old, her brother had pancreatic cancer at 43 years old, her maternal grandfather had colorectal cancer at 50+ years old and her maternal grandmother had gastric cancer at 50+ years of age. These familial cancer cases suggest a potential underlying genetic predisposition. In the context of colorectal cancer, a study conducted by Farzad Rahmani investigated the association between a specific single-nucleotide (SNP) polymorphism rs10811661 in the *CDKN2A/B* gene and the risk of colorectal cancer [35]. The study involved 541 individuals, both with and without cancer. The results demonstrate a significant association between the rs10811661 polymorphism and an increased risk of colorectal cancer. This suggests that individuals carrying this SNP may have an increased susceptibility to colorectal cancer [35]. In biliary tract cancers (BTCs), which encompass gallbladder cancer, intrahepatic cholangiocarcinoma (ICC), and extrahepatic cholangiocarcinoma (EHC), alterations in the *CDKN2A* gene are frequently observed. These alterations can occur through hypermethylation, homozygous deletion, or inactivating mutations. Specifically, studies have shown that *CDKN2A* alterations are more prevalent in GBC compared to ICC or EHC. Furthermore, these alterations have been associated with decreased survival rates in patients with BTCs [36,37].

The *CDKN2A* gene plays a significant role in the development of colorectal cancer and biliary tract cancer [35,36]. The presence of the rs10811661 polymorphism in the *CDKN2A/B* gene may increase the risk of colorectal cancer [35]. Additionally, alterations in *CDKN2A* are frequently observed in BTCs and are associated with poorer prognosis, particularly in GBC [36,37]. Understanding these genetic factors can inform risk assessments, screening strategies, and potential therapeutic approaches for individuals with a predisposition to these cancers.

Treatment of patients with the *CDKN2A* gene mutation variant does not have a specific targeted therapy. Nevertheless, considering the efficacy of CDK4/6 inhibitors in hormone-dependent, HER2-negative breast cancer, there is a hypothesis that these inhibitors may prove beneficial in cases where mutation variants in the *CDKN2A* gene conduce to overexpression of CDK4/6 proteins [38].

In preclinical studies involving pancreatic adenocarcinoma cell cultures with a *CDKN2A* gene mutation, CDK4/6 inhibitors demonstrated antiproliferative activity against these cancer cells [39,40,41]. In a prospective phase II clinical trial known as TAPUR, where 12 patients with pancreatic cancer and a variant mutation in the *CDKN2A* gene received Palbociclib monotherapy, no objective response was confirmed after 16 weeks. In this case, CDK4/6 inhibitor monotherapy was not effective [42]. However, another preclinical study reported the successful use of a combination of CDK4/6 inhibitors with MEK inhibitors and mTOR inhibitors [43,44].

The main challenge in treating pancreatic cancer patients is the extensive genetic heterogeneity and genomic adaptability of tumor cells, which enables the evasion of the targeted effects of most therapeutic drugs [45]. In addition, the stroma of pancreatic tissue is characterized by strong desmoplastic features that impede drug access to tumor cells. [46,47]. Undoubtedly, further research and advancements in treatment strategies targeting this mutation and activated proteins are imperative.

Similar to pancreatic cancer, monotherapy with CDK4/6 inhibitors did not lead to positive outcomes in the treatment of hereditary melanoma. On the contrary, studies by Zhang et al. demonstrated that blocking CDK4/6 kinases can stimulate PD-L1 expression, subsequently impairing the antitumor immune response and reducing the infiltration of lymphocytes into the tumor tissue [48]. However, the use of CDK4/6 inhibitors with immunotherapy targeting PD-1 receptors enhanced tumor regression and increased survival rates in mouse models of cancer [49].

While preclinical and clinical evidence suggests a potentially favorable impact of CDK4/6 inhibitors on this condition, further investigations are essential in the realm of personalized medicine for individuals with hereditary pancreatic cancer and melanoma. This ongoing research is vital for refining treatment approaches and improving outcomes for affected individuals.

## 5. Conclusions

This study contributes valuable insights into the genetic underpinnings of hereditary cancer syndromes, urging the continued exploration of personalized treatment approaches for individuals with CDKN2A mutations. The journey toward understanding and effectively managing hereditary pancreatic cancer and melanoma is ongoing, with the hope of improving outcomes and advancing the field of personalized medicine. Mutations in the *CDKN2A* gene present one of the most prevalent causes of increased susceptibility to pancreatic cancer. The potential founder effect of the identified mutations cannot be excluded, as the c.71G>C and c.159G>C variants were found in multiple cases. A larger sample of patients needs to be studied in order to determine their frequency in the population. The identification of individuals with a pathogenic *CDKN2A* variant holds significance for screening at-risk relatives and devising an effective clinical strategy for the patient.

## Figures and Tables

**Figure 1 biomedicines-11-03343-f001:**
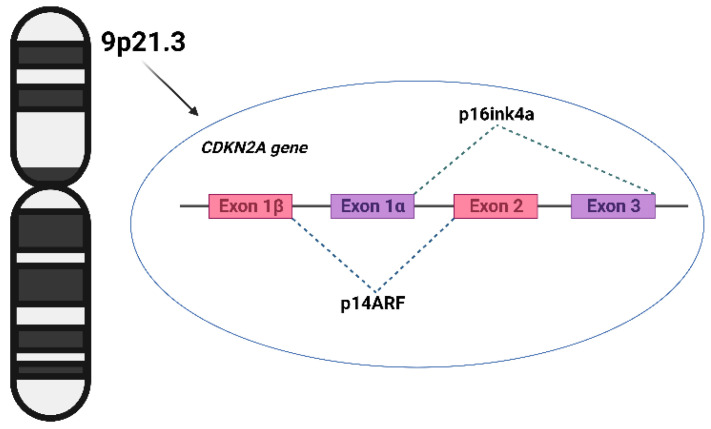
Structure of gene *CDKN2A.* The image was created using the Internet resource BioRender.com.

**Table 1 biomedicines-11-03343-t001:** Recommendations for monitoring patients with hereditary forms of pancreatic cancer and melanoma.

Melanoma	Pancreatic Cancer
Self-examination of the skin: Regularly check for the emergence of new nevi or any changes in the appearance of existing ones. Inspect existing nevi: keep a vigilant eye on existing nevi to monitor alterations in their shape, size, and color. Biannual dermatologist examinations: Schedule visits with a dermatologist twice a year. This includes the mapping of moles, followed by a dermatoscopy for a detailed examination. In case of suspected melanoma, consider the scraping of nevi followed by cytological examination. Sun protection for high-risk individuals: Individuals with familial melanoma and carrying pathogenic mutations in the *CDKN2A* gene, who have an elevated risk of melanoma, should regularly use sunscreens and take measures to avoid direct skin exposure to sunlight. Early screening for children in high-risk families: Initiate screening procedures and melanoma diagnosis for children in families with identified cases of melanoma from the age of 10.	Screening at age 40 (or 10 years earlier than earliest family diagnosis): Begin screening for pancreatic cancer at the age of 40, or 10 years prior to the earliest diagnosis of pancreatic cancer in the family history. Annual MRI with contrast: Undergo an MRI with contrast once a year, starting between the ages of 30 and 35. Endoscopic ultrasound examination: Consider an endoscopic ultrasound examination for a detailed assessment of the pancreas.

**Table 2 biomedicines-11-03343-t002:** Clinical characteristics and frequency of P/LP *CDKN2A* variants among tested individuals.

Patient №	Gender	Diagnosis	Stage	Histological Characteristics	Chromosomal Change	Coding	Protein	ACMG	Cancer Cases in Family History	GnomAD Exomes
1	F (42)	C24.0 Biliary tract cancer	IIB	Adenocarcinoma, G2	chr9:21971051delCG	c.307_308del	p.Arg103AlafsTer16	P	+	Not found
		C19 Colorectal cancer	I	Adenocarcinoma (KRAS, NRAS, BRAF negative)						
2	F (36)	D23.5 Dysplastic nevus		Dysplastic nevus, low grade	chr9:21974757C>G	c.71G>C	p.Arg24Pro	P	+	ƒ = 0.0000169
3	M (64)	C25.1 Malignant neoplasm: Body of pancreas.	III	Adenocarcinoma, G2	chr9:21974757C>G	c.71G>C	p.Arg24Pro	P	+	ƒ = 0.0000169
4	M (62)	C25.0 Malignant neoplasm: Head of pancreas.	IV	Adenocarcinoma, G2	chr9:21971051delCG	c.307_308del	p.Arg103AlafsTer16	P	+	Not found
5	F (41)	C25.0 Malignant neoplasm: Head of pancreas.	II	Neuroendocrine tumor, G1	chr9:21971200C>0	c.159G>C	p.Met53Ile	P	+	ƒ = 0.00000905
		C43.0 Melanoma	II	Melanoma						
6	F (51)	C25.0 Malignant neoplasm: Head of pancreas.	III	Adenocarcinoma, high grade	chr9:g.21971051delCG	c.307_308del	p.Arg103AlafsTer16	P	+	Not found

NM_000077.5 F—female, M—male, P—pathogenic. ACMG—American College of Medical Genetics and Genomics; GnomAD—The Genome Aggregation Database. Cancer cases in family history are presented in the Appendix A.

## Data Availability

The data are not publicly available since they contain information that could compromise the privacy of research participants. Requests to access the additional data should be addressed to the following email: s.nikolaev@mknc.ru.

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
