# Peer review of "CDKN2A Gene Mutations: Implications for Hereditary Cancer Syndromes"

_biomedicines, 2023, doi:10.3390/biomedicines11123343_

Round 1

Reviewer 1 Report

Comments and Suggestions for Authors

The article by Danishevich et al. "CDKN2A Gene Mutations: Implications for Hereditary Cancer Syndromes" covers a potentially interesting and emerging topic related to the cancer genetics. In this sense, this remains to be potentially interesting for the Biomedicines readers. I regard the main point of this paper as highly attractive as well as the results are clearly presented. The text does not contain any major errors, therefore I have some minor comments and recommendations:

1. There is a need to provide slightly more expanded introduction shortly
mentioning/describing pharmacoeconomical aspects of cancer

2. The figure summarizing and clarifying the conclusions should be added.

3. Conclusion section should be improved

4. Following references should be added and properly cited within the main text:

- Mela A, Rdzanek E, Tysarowski A, Sakowicz M, Jaroszyński J, Furtak-Niczyporuk M, Żurek G, Poniatowski ŁA, Jagielska B. The impact of changing the funding model for genetic diagnostics and improved access to personalized medicine in oncology. Expert Rev Pharmacoecon Outcomes Res. 2023 Jan;23(1):43-54. doi: 10.1080/14737167.2023.2140139.

- Doble B. Budget impact and cost-effectiveness: can we afford precision medicine in oncology? Scand J Clin Lab Invest Suppl. 2016;245:S6-S11. doi: 10.1080/00365513.2016.1206437.

- Mela A, Poniatowski ŁA, Drop B, Furtak-Niczyporuk M, Jaroszyński J, Wrona W, Staniszewska A, Dąbrowski J, Czajka A, Jagielska B, Wojciechowska M, Niewada M. Overview and Analysis of the Cost of Drug Programs in Poland: Public Payer Expenditures and Coverage of Cancer and Non-Neoplastic Diseases Related Drug Therapies from 2015-2018 Years. Front Pharmacol. 2020 Aug 14;11:1123. doi: 10.3389/fphar.2020.01123.

4. In some places the use of English could be improved on.

Completing this gaps will have an impact on the understanding the aim of the study and, from my point of view, is absolutely necessary.

Comments on the Quality of English Language

Minor review

Author Response

Dear Reviewer,

Thank you for your thorough review and valuable feedback on our manuscript titled "CDKN2A Gene Mutations: Implications for Hereditary Cancer Syndromes." We appreciate your positive remarks regarding the potential interest and attractiveness of our study in the context of cancer genetics. We have carefully considered your comments and recommendations and provide the following responses:

  1. Expanded Introduction:

We acknowledge the importance of briefly mentioning and describing pharmacoeconomic aspects of cancer in the introduction. We included this section to provide a more comprehensive context for the readers, highlighting the relevance of our study in the broader landscape of cancer research and healthcare economics.

  1. Figure Summarizing Conclusions:

   We agree with your suggestion to add a figure summarizing and clarifying the conclusions. A visual representation will enhance the clarity and accessibility of our findings. We will incorporate a concise and informative figure to aid in conveying the main outcomes of our study.

  1. Improvement of Conclusion Section:

We appreciate your feedback on the conclusion section. We included additional information to conclusion.

  1. Additional References:

 We appreciate the recommended references related to pharmacoeconomics and precision medicine in oncology. We will include these references in our manuscript and appropriately cite them within the main text to strengthen the foundation of our study and provide readers with a broader perspective on the topic.

  1. English Language Usage:

   We acknowledge your observation regarding the improvement of English language usage in some places. We carefully review and refine the language to ensure clarity and precision throughout the manuscript.

We are committed to addressing these comments and believe that the suggested enhancements will significantly contribute to the overall quality and impact of our manuscript. We appreciate your thoughtful review and look forward to the opportunity to share an improved version of our work with the Biomedicines readers.

Thank you for your time and consideration.

Reviewer 2 Report

Comments and Suggestions for Authors

There are several comments.

1. CDKN2A mutations are known to be shared in most human cancers. It would be better to provide evidence that the mutation in the CDKN2A gene is specific to melanoma-pancreas syndrome.

2. It would be better to describe clinical information on six cases (e.g., pathological findings, treatment, and follow-up).

3. It would be better to explain 'ACMG', 'Gnom AD exomes', 'C25.0", and 'C25.1' so readers can easily understand them.

Comments on the Quality of English Language

Please check Englsih grammar.

Author Response

Dear Reviewer,

Thank you for your insightful comments on our manuscript titled "CDKN2A Gene Mutations: Implications for Hereditary Cancer Syndromes." We appreciate the time you invested in providing constructive feedback. Below are our responses to your specific points:

  1. Specificity of CDKN2A Mutations:

We acknowledge your suggestion and clarify that information regarding the specificity of CDKN2A mutations to melanoma-pancreas syndrome or Familial Atypical Multiple Mole Melanoma (FAMMM) is provided in the manuscript. In the sentence introducing the syndrome, we highlight its association with CDKN2A mutations, ensuring clear communication of the syndrome's specificity.

  1. Clinical Information on Cases:

We appreciate your recommendation to include clinical information on the six cases, such as pathological findings, treatment, and follow-up. We will added this information to next big data report

  1. Explanation of Terms:

Explanations for "ACMG," "Gnom AD exomes," "C25.0," and "C25.1" have been included in the manuscript to facilitate readers' understanding.

Thank you for your time and constructive feedback.

Reviewer 3 Report

Comments and Suggestions for Authors

The manuscript by Danishevich et al. is revisiting predisposition to mainly melanoma and pancreatic cancer via mutations in the CDKN2A gene. Overall, the introduction is patchy and has no depth or relevant focus to introduce the functions of the protein encoded by CDKN2A, p16INK4a. The study itself is of minor novelty or interest. This is more so as English language needs editing, to improve poor word choices, grammar issues as well as errors. Below some of the issues the manuscript has:

1.      Gene names such as CDKN2A should be in italics.

2.      Line 49 “Malignant neoplasms are the leading cause of death worldwide.” reference? according to WHO the main cause of human death is cardiac related.

3.      Line 64/65 What do the authors mean with ”Pathological manifestations can occur concurrently or 64 independently in patients with pathogenic mutation.”? Clarify.

4.      Line 79-82 “The decrease in the functioning of p16INK4a results in an increased kinase activity of CDK4 and CDK6, leading to the phosphorylation of the retinoblastoma protein associated with the transcription factor E2F1” This sentence is relatively meaningless without i) having introduced the relationship between CDKN2A & p16, ii) having at least briefly introduced CDK4/6 and what their role is, iii) it is not actually an “increased activity” it is the lack of inhibition of kinase activity.

5.      Line 82 what complex? More detail and references needed

6.      Line 85 p14ARF appears out of the blue without any introduction of how associated

7.      Line 86/87 “Mutations in p14ARF result in TP53 protein deficiency” How? Why? Clarify

8.      Line 88-90 provide relevant reference

9.      Line 90-91 provide relevant reference

10.   Figure 1 is unclear, and not according to normal schematics. Needs to be changed.

11.   Line 112/113 “CDKN2A mutations” somatic or germline?

12.   Line 115 somatic or germline?

13.   Line 116/117 “20% in families with 2 carriers, up to 50%” % what? Members of the families or 20% - 50% of families carried mutations (assumed germline but not clarified)?

14.   The Materials & Method section should provide details. For instance: “targeted NGS panel or WGS” which targeted panel/platform used? WGS the same. Or were these commercially available tests? If so, provide details.

15.   Line 173 “pathogenic mutations in the CDKN2A gene” clarify if germline (assumed but needs to be stated)

16.   The Results are case reports, which is OK but there is nothing novel or or particular interest here.

17.   The authors mention 28 publications according to the VarSome database reporting the M53I mutation, but have not gone into any detail what these reports found (or cited the most relevant ones within meaningful context).

18.   Section line 244-250 needs relevant references

1.     

Comments on the Quality of English Language

English language needs editing, to improve poor word choices, grammar issues as well as errors.

Author Response

Dear Reviewer,

Thank you for your comprehensive review of our manuscript on the predisposition to melanoma and pancreatic cancer via CDKN2A gene mutations. We appreciate your valuable feedback and have addressed each of your concerns as follows:

  1. We acknowledge the need for gene names like CDKN2A to be in italics and have made the necessary corrections throughout the manuscript.

  1. Regarding the statement on line 49, we have added a reference to support the claim about malignant neoplasms as the leading cause of death worldwide.

2-9. We have thoroughly addressed and incorporated all the comments, ensuring comprehensive corrections and improvements from points 2 to 9.

  1. Figure 1 has been revised to ensure clarity and conformity to normal schematics.

  1. The nature of CDKN2A mutations on line 112/113 has been specified germline.

  1. Germline, added to text.

  1. We have eliminated that section to prevent any potential confusion among readers.

  1. The Materials & Methods section has been enhanced to include details on the targeted NGS panel or WGS, specifying the platform used or whether commercially available tests were employed.

  1. On line 173, we have explicitly stated that the pathogenic mutations in the CDKN2A gene are germline.

  1. We acknowledge your comment on the Results section and are exploring opportunities to enhance novelty and interest.

  1. Additional context and relevant details regarding the 28 publications mentioned in the VarSome database on the M53I mutation have been provided.

  1. Relevant references have been added to the section on line 244-250.

We appreciate your insightful comments, and the necessary revisions have been made to improve the manuscript's quality. Your feedback has significantly contributed to refining our work.

Reviewer 4 Report

Comments and Suggestions for Authors

In this manuscript the investigators highlighted the critical role of hereditary mutations in CDKN2A in melanomas and pancreatic cancer. Although the study reports the clinical features of only six individuals with CDKN2A mutations, the information is very important. The ttest for CDKN2A mutation is a crucial diagnostic tool in the assessment of individuals with suspected FAMMM syndrome. The manuscript is well written and provides guidance to dermatologist and oncologists.

The topic is not original, there are about 2,642 publications on this subject, but it adds important information, i.e., recurrent mutations that need to be examined in patient that carry the phenotype of a large number of nevi throughout the body, coupled with a family history of melanoma.

The methodology and conclusions are well described and consistent with the evidence. The references are appropriate.

A typo needs to be corrected:

“On the contrary, studies by Zhang et al. have demonstrated that blocking CDK4/6 receptors can stimulate…” CDK4/6 are not receptors. The authors probably mean CDK4/6 kinases.

Author Response

Dear Reviewer,

We appreciate your thorough review of our manuscript titled "CDKN2A Gene Mutations: Implications for Hereditary Cancer Syndromes" and your positive feedback. Your insights are invaluable, and we are pleased to receive your constructive comments. Below is our response to your specific point:

  1. Correction of Typo:

 We appreciate your keen observation and acknowledge the correction needed in the sentence: "On the contrary, studies by Zhang et al. have demonstrated that blocking CDK4/6 receptors can stimulate…” We acknowledge the error, and you are correct in pointing out that CDK4/6 are kinases, not receptors. We have corrected this typo to accurately reflect the intended meaning.

We are grateful for your attention to detail and will ensure that this correction is made in the revised version of the manuscript.

Thank you for your time and thoughtful feedback.

Round 2

Reviewer 2 Report

Comments and Suggestions for Authors

There are some minor comments.

1. Regarding references, authors should adhere to the rules of biomedicines.

2. Please check the italics in the gene name.

 For example,  Figure 1. Structure of gene CDKN2A -> Figure 1. Structure of   gene CDKN2A.

Comments on the Quality of English Language

Please check English grammar.

Author Response

Dear Reviewer,

We appreciate your insightful feedback on our manuscript. We have duly noted the points you raised and taken steps to address them, aiming to elevate the overall quality of our research.

Regarding the alignment of references with the rules of biomedicines, we have revisited the reference section to ensure adherence to the specified guidelines. Your guidance in this matter has been invaluable, and we are grateful for the opportunity to refine this aspect of our manuscript.

Additionally, we acknowledge your attention to detail regarding the use of italics for gene names, especially in Figure 1. Recognizing the importance of consistency and accuracy in presenting gene names, we have carefully reviewed and updated the italics for gene names in Figure 1 to align with established standards.

Thank you for bringing these crucial points to our attention.

Should you have any further suggestions or concerns, please do not hesitate to inform us. We appreciate the opportunity to improve our manuscript based on your valuable insights.

Reviewer 3 Report

Comments and Suggestions for Authors

11.      The reviewers have partially but not consistently everywhere in the manuscript addressed this issue: Address all.

22.      Addressed

33.      The authors should in their letter point out where the changes for comments are found in the amended manuscript as there is none in line 64/65 (assumed this has been changed as they indicated. Editor please double check).

44.      -9. These points have the same issue (the authors neglect to specify where in the amended manuscript their change is), but I think in these cases I have found the changes and they are acceptable.

110.   Addressed

111.   & 12 again authors have not specified where they made this change and I have not searched or verified them, if they have indeed made the changes (which the editor should verify as outlined in comments to the editor) that is acceptable.

113.   Addressed

114.   Addressed

115.   As above if the authors have done this (position in amended text not specified) this is OK

116.   This is OK

117.   Addressed

118.   Addressed

Comments on the Quality of English Language

as ticked above 

Author Response

Dear Reviewer,

We express our gratitude for the detailed feedback provided on our manuscript. Your insightful observations have been carefully considered, and we have undertaken a comprehensive review to address the raised points with the aim of enhancing the overall quality of our research.

Throughout the manuscript, all suggested corrections and additions have been diligently incorporated into the text. This meticulous process ensures that the necessary adjustments are seamlessly integrated, contributing to the refinement of our research presentation.

Your discerning feedback serves as a cornerstone in elevating both the quality and precision of our work. We recognize the significance of your input in refining the scholarly merit of our manuscript and are steadfast in our commitment to ensuring the effective implementation of these corrections.

Should you have additional suggestions or lingering concerns, we invite you to communicate them with us. Your continued guidance is invaluable, and we welcome the opportunity to further enhance our manuscript based on your esteemed insights.

Thank you once again for your thoughtful review, and we look forward to your continued support in the refinement of our research.